# Nanomedicine and Immunotherapy: A Step Further towards Precision Medicine for Glioblastoma

**DOI:** 10.3390/molecules25030490

**Published:** 2020-01-23

**Authors:** Neja Šamec, Alja Zottel, Alja Videtič Paska, Ivana Jovčevska

**Affiliations:** Medical Centre for Molecular Biology, Institute of Biochemistry, Faculty of Medicine, University of Ljubljana, 1000 Ljubljana, Slovenia; neja.samec@mf.uni-lj.si (N.Š.); alja.zottel@mf.uni-lj.si (A.Z.)

**Keywords:** glioblastoma, cell free DNA, circulating tumor DNA, liquid biopsy, immunotherapy, extracellular vesicles, nanoscience

## Abstract

Owing to the advancement of technology combined with our deeper knowledge of human nature and diseases, we are able to move towards precision medicine, where patients are treated at the individual level in concordance with their genetic profiles. Lately, the integration of nanoparticles in biotechnology and their applications in medicine has allowed us to diagnose and treat disease better and more precisely. As a model disease, we used a grade IV malignant brain tumor (glioblastoma). Significant improvements in diagnosis were achieved with the application of fluorescent nanoparticles for intraoperative magnetic resonance imaging (MRI), allowing for improved tumor cell visibility and increasing the extent of the surgical resection, leading to better patient response. Fluorescent probes can be engineered to be activated through different molecular pathways, which will open the path to individualized glioblastoma diagnosis, monitoring, and treatment. Nanoparticles are also extensively studied as nanovehicles for targeted delivery and more controlled medication release, and some nanomedicines are already in early phases of clinical trials. Moreover, sampling biological fluids will give new insights into glioblastoma pathogenesis due to the presence of extracellular vesicles, circulating tumor cells, and circulating tumor DNA. As current glioblastoma therapy does not provide good quality of life for patients, other approaches such as immunotherapy are explored. To conclude, we reason that development of personalized therapies based on a patient’s genetic signature combined with pharmacogenomics and immunogenomic information will significantly change the outcome of glioblastoma patients.

## 1. Modern Medical Approaches for Cancer Treatment

In contrast to classical medicine, which uses universal treatment for all individuals affected by a disease, the goal of precision medicine is tailoring the treatment to an individuals’ needs [1]. To satisfy the goal of “right drug, right person, right time”, precision medicine will have to integrate genomic, proteomic, pharmacogenomic, and immunogenomic patient information so that the treatment will be most effective for the particular individual [2]. Development of this approach is time consuming and expensive. An important obstacle that appears in precision medicine is targeted delivery of drugs, which is a result of the existence of biological barriers such as the blood–brain barrier (BBB), and also interactions between nanoagents and the biological molecules they encounter on the way [1].

In general, the study of structures and molecules ranging from 1 nm to 100 nm is called nanoscience, whereas the study of its practical applications is called nanotechnology. Nanoscience refers to the manipulation of materials at atomic and molecular scales, and nanotechnology is the ability to manipulate, observe, measure, and manufacture matter at the nanometer scale. Nanotechnology is one of the most auspicious technologies of our era. Different nanomaterials including polymeric particles, micelles, nanoshells quantum dots and magnetic iron oxide nanoparticles have been tested for clinical application in brain tumor diagnosis [3]. In this regard, magnetic nanoparticles are among the most extensively explored nanomaterials. Magnetic nanoparticles are composed of an iron oxide core and biocompatible material, which can be polysaccharides, synthetic polymers, lipids, or proteins [3]. In order to be suitable for imaging and targeting, they should be coated with non-toxic and biocompatible neutral or negative surface coating.

### Molecular Pathology of Glioblastoma

Malignant brain tumors are a significant challenge for the medical system, as well as for patients and their families. Among gliomas, the grade IV glioblastoma is the most common and presents with discouraging prognosis. Despite the great technological advances in imaging, surgery, and adjuvant therapies, median survival is 14–16 months after diagnosis [4], while the 5-year overall survival is only 9.8% [3,4]. The infiltrative nature is one of the main reasons for tumor recurrence. Glioblastoma is a highly heterogeneous tumor, the diversity of which is presented at cellular [5,6] and subcellular levels [7,8,9]. Glioblastoma is composed of numerous cells (mature cells as well as glioblastoma stem cells) with different genetic properties, which is illustrated by the identification of the three main subtypes (mesenchymal, proneural, and classical [10]) within the same tumor.

## 2. Nanomedicines in Glioblastoma Diagnosis

There are a few widely used imaging methods for disease diagnosis that include computed tomography (CT), magnetic resonance imaging (MRI), positron emission tomography (PET), single-photon emission computed tomography (SPECT), and ultrasound imaging. Currently, glioblastoma is diagnosed with CT or MRI and magnetic resonance spectroscopy (MRS). Still, these methods are not able to detect small tumor portions that invade surrounding tissue or individual cells [11]. After obtaining a biopsy sample, this is evaluated and histopathologically processed, then tested for different biomarkers (e.g., isocitrate dehydrogenase 1/2 (IDH1/2) mutations and O-6-methylguanine-DNA methyltransferase (MGMT) methylation status) with a number of genetic and molecular techniques, such as fluorescence *in situ* hybridization (FISH) [12,13,14]. With the addition of genomic information into clinical diagnosis, the era of precision medicine was started. Diagnosing glioblastoma *in vivo* is complicated due to the existence of the protective semipermeable membrane known as the blood–brain barrier (BBB). However, nanoparticles are thought to pass the BBB through receptor-mediated endocytosis. For this purpose, nanoparticles ought to be coated with surfactants, which will allow specific adsorption of serum proteins, or should be attached to peptides or ligands for specific endothelial receptors [1]. A major concern is the appearance of neurotoxicity from the application of nanoparticles. To avoid unwanted side effects and potential damage, the metabolism, decomposition, and removal of nanoparticles from the brain should be thoroughly evaluated before their clinical application.

### 2.1. Nanoparticles

Development of nanoparticles as contrast agents to be used in imaging techniques allowed for information about the extent of the surgical removal to be obtained and also for specific drug delivery to tumor areas to be monitored [15]. The possible application of different nanoparticles for use as imaging agents for glioblastoma diagnosis has been tested *in vitro*, *in vivo*, and with human subjects, as summarized in Table 1 and explained below. Among these are multifunctional iron oxide nanoparticles (IONP), which have been explored for use as imaging agents for so-called molecular MRI [11]. The magnetic properties of iron oxide nanoparticles allow for their direct imaging in MRI [16]. In addition, they offer the possibility of attaching tumor-specific biomolecules to their biocompatible surface [17]. In order to additionally increase specificity, nanoparticles can be coated with polyethylene glycol (PEG). In the study by Hadjipanayis et al., the authors tested the epidermal growth factor receptor variant III (EGFRvIII) antibody–IONP complex *in vitro* and showed MRI contrast enhancement [18]. On the other hand, ultrasmall superparamagnetic iron-oxide-based nanoparticles show advantages over gadolinium-based MRI contrast agents, as they are eliminated more slowly, reside longer in tumor cells, and imaging can be performed 24 h to 72 h after administration [19]. Molecular MRI uses cell-specific proteins for targeted contrast agents composed of superparamagnetic nanoparticles binding to specific cellular targets [11]. Tomanek et al. developed a diagnostic method composed of IONP with infrared core functionalized with single-domain antibody targeted against the insulin-like growth factor binding protein 7 (IGFBP7) [11]. Using murine models, the authors showed that binding of the functionalized nanoparticles was not a result of passive accumulation, but through specific binding to the target IGFBP7, where the nanoparticles stay bounded for up to 24 h. The study also proved successful conjugation of nanoparticles for specific targeting of biomolecules and increased MRI specificity. These results can be implemented for therapeutic purposes by enhancing visualization on preoperative or intraoperative MRI, where fluorescing tumor vessels can be used to increase the extent of surgical resection.

### 2.2. Fluorescent Magnetic Resonance Imaging

In glioblastoma diagnosis, MRI is the most widely used method for providing anatomical information and images, but it has its limitations in terms of giving false positive signals due to macrophage engulfment of imaging vesicles, small hemorrhages, and iron depositions in the aging brain. Preoperative contrast-enhanced MRI using small molecules such as gadolinium is used to visualize BBB disruptions and determine the macroscopic outlines of the tumor [23]. Still, limited spatial resolution and differences between preoperative MRI and actual tumor borders during surgery are observed. On the other hand, intraoperative MRI is expensive, time consuming, and requires repeated gadolinium injections, which leads to surgically induced post-operative false positive contrast enhancement [23].

Another type of intraoperative imaging is by using fluorescence, which offers the benefits of high contrast, sensitivity, low cost, and by using appropriate microscope, visualization of cells and tissue in vivo and in vitro [27]. Different fluorescent probes are available for this purpose, such as “always on” low molecular weight probes with folate-receptor targeted fluorophore; “smart” probes, which are activated at the tumor site and are intact elsewhere; and “turn on” probes. such as fluorescently tagged activatable cell-penetrating peptides, low molecular weight fluorogenic peptides, and trigger-activated analytes [28,29,30,31]. It is recommended that a fluorescent imaging probe be selective towards the tumor cells while sparing adjacent tissue. An example of this is the application of the Food and Drug Administration (FDA)-approved metabolic precursor of fluorescent porphyrin IX (PpIX), named 5-aminolevulinic acid (5-ALA), for intraoperative MRI, with a recommended dose of 20 mg/kg body weight [21]. The 5-ALA accumulates in non-necrotic marginal glioblastoma tissue [32]. It is metabolized in the mitochondria of glioma cells, where it produces protoporphyrin IX, which fluoresces under deep blue light, allowing for the immediate identification of tumor tissue [21,22]. In the study by Ji et al., the authors performed analysis of the fluorescence-guided surgery in numerous gliomas, while focusing on epidemiological data of the fluorescence pattern and its impact on extent of resection [20]. They reported that 95.4% of glioblastomas showed positive fluorescence, with 85.6% giving strong signal; while grade III, II, and I gliomas showed positivity in 55%, 24.1%, and 26.3% of the cases, respectively. In addition, complete resection was performed in 89.6% fluorescence-positive gliomas compared to 75% of fluorescence-negative gliomas. In a retrospective cohort study, Michael et al. tested the effect of different doses of 5-ALA on residual tumor volume and patient overall survival [21]. However, the authors did not find a correlation between either low (10–30 mg/kg) or high (40–50 mg/kg) doses and overall survival. Still, patients who received high dosage of 5-ALA were associated with less residual tumor volume and were more likely to exhibit gross total resection [21].

Today, 5-ALA-guided surgery requires expensive equipment (i.e., neurosurgical microscopes with blue light modules), making it challenging for developing countries. For this purpose, Woo et al. developed a wavelength-specific blue and white light-emitting diode (LED) headlamp for glioblastoma resection [22]. The authors illustrate the usefulness of this headlamp using three glioblastoma cases. In one of the patients, the headlamp illuminated tumor areas where fluorescence was not observed with the Leica operating microscope. In the other cases, fluorescence was comparable. Since the extent of the surgical resection is an important therapeutic predictor for glioblastoma patients, it is desired for the surgery to be as extensive as possible without causing additional damage. The headlamp the authors are proposing possesses two main advantages: first, it offers the possibility to change between blue and white light without an external laser; and second, it allows for greater freedom of movement for the neurosurgeon. At last, the cost of constructing and maintaining such a device is considerably lower compared to these costs for microscopes. Still, the authors of this study do not suggest their headlamp should replace microscopy-based surgery, but that it can assist in it. However, in developing countries with limited resources, such a low cost device can enable surgeons to perform fluorescence-guided resections where 5-ALA imaging is unavailable.

The use of 5-ALA is convenient as it provides real-time information about the tumor borders, which leads to increasing the extent of the surgical resection in tumor infiltrated areas, and ultimately to increased overall patient survival [20]. However, problems with autofluorescence of non-tumor tissue (insufficient specificity) and subjectivity in the interpretation of results are often observed [33]. Other factors that should be considered are tumor necrosis, which lacks fluorescence; hypervascularity and its related hemorrhage; patient body weight; and adjusting the surgery within the time window of maximum fluorescence intensity (4 h to 9 h after oral administration) [20]. However, an advantage of the use of fluorescent probes for imaging is that they can be engineered to be activated through specific molecular pathways. This will lead to personalization of the imaging methodology according to the patient’s specific biomarkers [27].

### 2.3. Other Imaging Methods

Raman scattering is a non-destructive method that provides information about the molecular composition of the sample and its structure [34]. Raman spectroscopy shows limited applications for in vivo imaging due to its poor signal-to-noise ratio and long acquisition times [24]. Gold nanoparticles can enhance the Raman signal because of their plasmonic effect, which is named the “surface-enhanced Raman spectroscopy” (SERS) phenomenon. Compared to intrinsic Raman spectroscopy, SERS shows enhancement of several orders of magnitude [35]. Another improvement is the implementation of a Raman reporter with an electronic transition similar to that of the laser excitation wavelength, or so-called “surface enhanced resonance Raman spectroscopy” (SERRS) [36].

The combination of Raman scattering (SERRS) nanostars resonant in the near-infrared (NIR) wavelength range [37,38] or SERRS combined with multispectral optoacoustic tomography (MSOT) [23] is another approach for glioblastoma imaging. The dual SERRS-MSOT in vivo imaging reported by Neuschmelting et al. can detect the true microscopic extent of a tumor (loosely scattered tumor cells), especially at the tumor margins, which cannot be detected by naked eye and are a major cause for tumor recurrence [23]. Despite their sensitivity towards glioblastoma and glioblastoma-periphery, the SERRS-MSOT nanostars are not clinically approved yet due to concerns about neurotoxicity.

A relatively new technique named “spatially offset Raman spectroscopy” (SORS) aims to overcome the limitations of classical Raman spectroscopy [24]. In general, Raman spectroscopy relies on inelastic light scattering, and SORS utilizes the concept of random scattering, in which deeper born Raman photons are less likely to travel back to the initial source. Therefore, in turbid media, these photons will have to travel more than shallow photons. Combining the depth penetration benefits of SORS with the signal enhancing capabilities of SERS led to the “surface-enhanced spatially offset Raman spectroscopy” (SESORS) method for examination of samples at greater depths [24]. Using murine models, Nicolson et al. examined the potential of SESORS for in vivo imaging. The authors used Raman-active SERRS nanotags with cyclic arginine–glycine aspartic acid (cRGD) tripeptide-targeted against integrin receptors that were tracked using “surface enhanced spatially offset resonance Raman spectroscopy” (SESORRS). In the study, the authors showed non-invasive visualization of glioblastoma in murine models. So far, this method has been tested on experimental animals only, so its clinical implementation is yet to be determined.

In the study by Cho et al., the authors tested the physico-chemical and magnetic properties of nanocubes (NCs) and assembled larger nanocube constructs (ANCs) coated with double oleic acid (dOA) or bovine serum albumin (BSA) [25]. The in vitro cytotoxicity testing of these constructs using U87MG human glioblastoma cells and J774 murine macrophages did not show a toxic effect of the dOA-ANCs or BSA-ANCs on the cells. BSA-ANCs were further used for in vivo studies on murine models of glioblastoma due to their longer stability over time. Although a signal was observed in liver, spleen, and kidneys, assessing the biodistribution of BSA-ANCs showed selectivity towards the malignant mass. The authors reported that BSA-ANCs are also suitable for in vivo MRI performance, as they specifically accumulate in the tumor site. The authors suggest that the use of NCs is a valuable strategy for cancer detection and therapy.

A novel method for in vivo tracking of glioblastoma is the single-photon emission-computed tomography (SPECT) imaging with actively migrating neural stem cells developed by Cheng et al. [26]. The authors validated in vivo tracking of ^111^In-mesoporous silica nanoparticle (MSN)-labeled neural stem cells migrating towards a glioma xenograft. The sensitivity and spatial resolution of this system can be further improved using advanced imaging equipment. An advantage of MSN is their large surface area, which allows for efficient loading with selected drugs. This system allows for further investigation of the behavior of therapeutic stem cells.

In general, advances in material engineering and the development of nanotechnology has the potential to revolutionize the way we diagnose and treat glioblastoma [15]. However, development of powerful and sensitive techniques, such as next generation sequencing, will allow the detection of genomic information in biological fluids such as urine, saliva, blood, and cerebrospinal fluid [2]. This non-invasive diagnostic approach for disease detection and monitoring of disease recurrence is called “liquid biopsy”.

## 3. Liquid Biopsies

Therapeutic response and recurrence after therapy are usually monitored with MRI, which does not distinguish tumor progression from radiation necrosis [39]. Current MRI has a resolution limit of approximately 2–3 mm. This means that tumor cells will have to undergo more than 20 rounds of cell divisions to be detected by MRI [40]. The biopsy process itself confers a non-negligible degree of risk related to the procedure. For example, there may be a considerable brain swelling within and around the tumor itself; hemorrhage that occurs in the biopsy tract could be potentially threatening to the patients’ neurological function or even life. Biopsies in the brain are very challenging from the perspective of obtaining samples or repeated prospective collection of tissue material [41,42], and may not fully capture the intratumoral heterogeneity [42]. There is a great need to obtain new tools to determine treatment response, molecularly characterize tumor profiles, and to minimize the aggressiveness associated with surgery [41].

In contrast to traditional tissue biopsies, liquid biopsy can be repeated regularly with minimal risk and can serve as a powerful approach to assess dynamic, real-time molecular information from tumors [43]. Blood, cerebrospinal fluid (CSF), and urine samples contain tumor components in the form of extracellular vesicles (EVs), as well as circulating tumor DNA (ctDNA) and circulating tumor cells (CTC), as presented in Figure 1. Their functions contribute to sophisticated communication between several cell types surrounding tumors [39]. Liquid biopsy is a non-invasive method with which relevant information is obtained quickly and inexpensively [44]. Analysis of CTC, EVs, and ctDNA can potentially characterize the global tumor genome and transcriptome [42]. The advantages are in obtaining far more balanced molecular profiles of the tumors, and the process can be performed multiple times during the clinical therapy, which could prevent molecular sub-typing of the tumor cells [42,44]. There are three key considerations for the selection of suitable candidate biomarkers for glioma liquid biopsy. These are ease of access, stability, and detection sensitivity, which is especially relevant to glioblastomas, where the highly dynamic nature of the tumor can lead to clinically relevant changes within months or even weeks [43]. Tissue biopsy of glioma samples may not reflect an entire tumor molecular profile because glioma tumors are known for their heterogeneity [41]. In fact, 60%–70% of mutations identified in tumors were not present in all regions of the tumor [45]. Analysis of tumors with tissue biopsy limits the analysis to a single point in time, whereas tumors evolve in response to treatment [46]. Liquid biopsy of blood and CSF in conjunction with tissue biopsy offers opportunities to confirm diagnosis of present mutations and monitor tumor evolution and response to therapy [47]. The ability to monitor tumor changes, ideally before they are clinically or radiologically apparent, would significantly improve the clinical management of glioma patients.

### 3.1. Extracellular Vesicles

Cell-secreted vesicles with sizes in the range of 30–2000 nm are called extracellular vesicles (EVs) and are involved in cellular remodeling, intracellular communication, modulation of the tumor microenvironment, and regulation of immune function [48]. Extracellular vesicles are secreted by most (if not all) cell types, and reflect the identity and molecular state of their cell-of-origin [49]. EVs are divided into three main classes of secreted vesicles, endosome-derived exosomes (30–100 nm), plasma-membrane-derived microparticles (100–1000 nm), and apoptotic cell-derived apoptotic bodies (1000–5000 nm), all of which vary in size, content, and biogenesis [48,50]. Exosomes form through intraluminal invagination of the membrane of late endosomes to form multivesicular bodies, which then fuse with the plasma membrane to secrete the intraluminal vesicles as exosomes [50]. Exosomes are also being recognized as promising candidates for liquid biopsy development. The physical and biochemical properties of exosomes make them exploitable as stable biomarker reservoirs, and are relatively stable in blood, with reports of a half-life of up to 5 h in patients with thrombocytopenia [43]. Exosomes secreted by tumor cells can pass anatomic compartments and are detected in biofluids, such as blood, CSF, and other biofluids from cancer patients [51,52]. They carry constellations of molecules (RNA, DNA, proteins, and lipids) enclosed within a lipid bilayer and protected from enzymatic degradation. Secreted tumor EVs enable remodeling of the extracellular matrix and possess matrix-degrading metalloproteases (MMP), such as membrane type-1 MMP (MTI-MMP), MMP9, MMP2, and urokinase-type plasminogen activator [53,54]. Skog et al. performed the first comprehensive study on the profile of the molecular contents of glioma exosomes. A total of 27,000 enriched transcripts were characteristic of glioma, including transcripts encoding epidermal growth factor receptor variant III (EGFRvIII) and glial fibrillary acidic protein (GFAP), within exosomes derived from glioblastoma cells in vitro [51]. Moreover, glioma tumor cells secrete exosomes in significantly greater quantities than normal cells, allowing tumor-derived molecules to be detected above the levels of potential noise from EVs released from other sources [43].

EVs isolated from glioblastoma cell lines contain tumor-specific micro RNA (miRNA) and messenger RNA (mRNA), which can also be detected in exosomes derived from the biofluid of glioblastoma patients. Tracking of these biomarkers from the patient biofluid may serve in identification of therapeutic response and recurrence of disease [55]. Manda et al. established a robust method incorporating two different primer sets to detect exosome *EGFRvIII* through serum exosomes. They could only detect *EGFRvIII* RNA and not wild-type epidermal growth factor receptor (*wtEGFR*) RNA [55], which is in line with previous findings that *wtEGFR* is undetectable in exosomes due to the larger size of the transcript [56]. The accuracy of *EGFRvIII* detection through exosomes was 80% for tissue *EGFR* expression, with an overall sensitivity and specificity of 81.58% and 79.31%, respectively [55]. Figueroa et al. obtained CSF shortly after resection of the primary glioblastoma, where *EGFRvIII*-positive, CSF-derived EVs had significantly more *wtEGFR* RNA expression. *EGFRvIII* was detected in CSF-derived EVs for 14 of 23 *EGFRvIII* tissue-positive glioblastoma patients. Results showed a sensitivity of 61% and specificity of 98% for the ability of CSF-derived EVs to detect an *EGFRvIII*-positive glioblastomas [57]. The ability of sensitive and specific analyses of *EGFRvIII* in EV-derived RNA from the CSF might be obtained from lumbar puncture at the time of MRI detection of intracranial mass, given that this procedure is relatively safe when appropriate precautions are taken in patients with increased intracranial pressure [58]. Chandran et al. identified syndecan-1 (SDC1) as a plasma EV constituent that discriminates between high-grade glioblastoma (World Health Organization (WHO) grade IV), low-grade glioma (LGG, WHO grade II), and plasma EV SDC1 correlated with SDC1 protein expression in matched patient tumors, for which the level of plasma EV SDC1 was decreased after surgery [59]. Mutations in IDH1 are found in 10% of all gliomas and 80% of secondary gliomas [60]. The majority of IDH1 mutations consist of arginine to histidine mutation at NA 132 [61], after the which the enzyme becomes deficient but harbors new catalytic activity, namely the production of 2-hydroxyl-glutarate, which is associated with an altered chromatin state [62]. The mutant transcript of *IDH1* can be detected in EV isolated from CSF, and the quantity of mutant *IDH1* transcripts is directly correlated with the tumor volume [63]. The miRNA signatures for *IDH*-wt glioblastoma and *IDH*-mutant grades II and III gliomas have been described in serum exosomes and could distinguish preoperative glioblastoma patients from healthy controls with high accuracy [64]. The protein miR-21 is a miRNA that is highly overexpressed in glioblastoma tissue cells [65]; miR-21 levels in exosomes isolated from the CSF of glioblastoma patients were 10-fold higher than those from controls. Akers et al. were able to distinguish CSF derived from glioblastoma patients and non-oncologic patients; additionally, miR-21 in CSF EVs decreased after surgical resection of tumor, and the authors could follow-up their levels of miR-21 [52]. Several studies also included serum-derived EVs, where miR-21 was reported as significantly upregulated [66]. Moreover, miR-320, miR547-3p, and RNU6-1 were significantly associated with glioblastoma diagnosis and overall survival [67]. Levels of serum EV-associated miR-301 have also been found to be elevated in glioma patients by quantitative real-time PCR, the levels of which fall after surgery and are predictive of overall survival [68].

### 3.2. Circulating Tumor Cells

Circulating tumor cells (CTCs) are cells that have left the primary tumor and entered the blood flow. It is not known yet if they are a subpopulation of the central tumor or if they leave tumor site randomly, thus representing the entire original tumor. Regardless, CTCs exist in the blood of metastatic patients [42,69,70]. Recent actions in this field have led to the capture, identification, and analysis of CTCs [71]. Their abundance in the bloodstream is very limited at only 1 CTC per 10^9^ blood cells, which is also the reason that their difficult isolation requires complex techniques [72]. CTCs are isolated using antibody-mediated capture by targeting extracellular membrane proteins. For non-epithelial tumors such as brain gliomas, membrane-specific protein identification is extremely difficult. Only those within the target will be isolated using this approach [42]. Another positive selection strategy for an abundance of specific tumor markers that are commonly expressed on the surface of cells is to isolate them with differential centrifugation followed by fluorescence immunocytochemistry, using GFAP as a marker for CTCs [73]. Macarthur et al. detected glioblastoma CTCs using nestin and human telomerase markers [71]. Other methods sought to enrich CTCs by removing the unwanted components of whole blood. CTCs can be isolated due to their size because CTCs are larger than normal blood cells, which is named the so-called negative enrichment approach. Sullivan et al. isolated CTCs with CTC isolation chip (iChip) technology, where a combination of microfluidic flow manipulation and hydrodynamic flow sorting was used to remove small parts in blood, inertial flow was used to align nucleated cells, and magnethophoresis was used to remove white blood cells [7]. In another negative selection approach, glioblastoma CTCs were isolated with erythrocyte lysis and density gradient separation, along with removal of leukocytes using magnetic beads [42,74]. CTCs have clinical utility as a biomarker of prognosis (71% of glioblastoma patients contain CTCs) and might be useful as a therapeutic response. CTCs were detected in different grades of glioma and they represent the tumor profile [75]. Glioblastoma-derived CTCs presented *EGFR* amplification, which is linked with the presence of *EGFRvIII* and aggressiveness of the disease [74]. Cultured CTCs were shown to express the sex-determining region Y-box 2 (SOX2), octamer-binding transcription factor 4 (OCT4), and homeobox protein NANOG, which are glioblastoma stem cell markers [76], and also markers associated with very aggressive mesenchymal subtype, serpin family E member 1 (SERPINE1), vimentin (VIM), transforming growth factor-beta 1 (TGFB1), and transforming growth factor-beta (TGF-β) receptor type 2 (TGFBR2) [7]. They also contribute to resistance to treatment with temozolomide and radiotherapy [76].

### 3.3. Cell Free Circulating Tumor DNA

Mandel and Métais detected and quantified the presence of cell-free nucleic acid in 1948 in human blood from healthy and diseased patients for the first time [77,78]. Stroun et al. proved the presence of neoplastic characteristic plasma DNA in 1989 [79]. Vasioukhin and Sorenson in two different studies confirmed that tumors can shed DNA into the circulation [80,81]. Circulating tumor DNA or ctDNA can be released from CTCs, primary tumors, and secondary tumors into the circulation of cancer patients, and can be bound to complex proteins, cell surfaces, or vesicles [82]. Different kinds of tumor-specific DNA aberrations may be recorded in cfDNA, such as point mutations, loss of heterozygosity (LOH; gene amplifications; presence of viral oncogenic DNA; hypermethylation of tumor suppressor gene promoter areas; and hypomethylation of long, interspersed nucleotide element-1 [83,84,85]. The ctDNA is more fragmented than normal cfDNA [86] and it can exist at sizes less than 1000 bp [42]. Typically, there is a very low concentration of nucleic acid in biofluids compared to cells or tissue (less than 100 ng/mL in plasma), thus leaving only a few molecules per sample to be detected. This challenge requires optimized methods for DNA extraction. Higher sample volumes permit isolation of sufficient absolute amounts of ctDNA for assay, low elution volume is required for higher concentrations in the eluate after purification, and there is a need to efficiently recover smaller fragments. However, it is possible to detect tumor-associated nucleic acid in various biofluids in brain tumor patients [42]. However, the fraction of ctDNA in cancer patients may vary, accounting for 3–93% of the total cfDNA [87]. The cfDNA of 8 glioblastoma patients fluctuated during treatment, with the highest levels seen before surgery and at progression. An increase was seen in all patients at the time of progression, while no increase was seen in 3 out of 4 patients without progression. There was a good tendency shown between cfDNA of treatment course and response [88]. Circulating blood-based biomarkers are considered to be of great need for diagnosis, molecular characterization, and treatment response determination for gliomas [41]. Circulating tumor DNA was found in the plasma of 10% glioma patients in one study [89], whilst it was detectible in 55% of patients sera using methylation assays in a different study [43,90]. Schwaederle et al. report the results of liquid biopsies in 171 patients with a variety of cancers, whose blood was analyzed for 54 genes via next generation sequencing (NGS) in circulating tumor cell-free DNA. Overall, 58% patients had at least one detectable alteration, most commonly in tumor protein p53 (*TP53,* 29.8%), *EGFR* (17.5%), tyrosine-protein kinase Met (*MET,* 10.5%), phosphatidylinositol-4,5-bisphosphate 3-kinase catalytic subunit alpha (*PIK3CA,* 7%), and Notch receptor 1 (*NOTCH1,* 5.8%). In total, 27% of glioblastoma patients had at least one detectable alteration, most commonly *TP53* and *NOTCH1* anomalies [91]. Study from Zill et al. showed 51% ctDNA detection rate from cfDNA in patients with advanced primary glioblastoma [92]. Piccioni et al. studied ctDNA in 419 patients with glioblastoma and other primary brain tumors. Half of primary brain tumor patients that had detectable ctDNA had genomically targetable off-label or clinical trial options, which is contrary to previous studies with very low yields. A total of 211 (50%) patients had 1 or > 1 somatic alterations detected. Detection was highest in meningioma (59%) and glioblastoma (50%); single nucleotide variants were detected in 61 genes (most frequent in *TP53* (*n* = 79), followed by Janus kinase 2 (*JAK2*, *n* = 10), neurofibromatosis type 1 (*NF1*, *n* = 7), *EGFR* (*n* = 7), proto-oncogene B-Raf (*BRAF*, *n* = 6), *IDH1* (*n* = 5), neuroblastoma RAS viral oncogene homolog (*NRAS, n* = 5), guanine nucleotide-binding protein (G protein), alpha-stimulating activity polypeptide 1 (*GNAS*, *n* = 5), and ataxia-telangiectasia mutated gene (*ATM*, *n* = 4). Amplifications were detected in Erb-B2 receptor tyrosine kinase 2 (*ERBB2*), *MET*, *EGFR*, and others [93]. CSF ctDNA is enriched in brain tumors and produces better results than plasma ctDNA [94]. In CSF ctDNA drug resistance, mutations were identified in patients whose CNS disease progressed during kinase inhibitor therapy [95] and telomerase reverse transcriptase (*TERT*) promoter mutations [96]. The sub-classification of diffuse glioma is managed through CSF ctDNA platforms, which simultaneously test seven genes: *IDH1*, *IDH2*, *TP53*, ATP-dependent helicase ATRX (*ATRX*), *TERT*, histone H3.3 (*H3F3A*), and histone H3.1 (*HIST1H3B*) [97]. Miller et al. showed that tumor-derived DNA was detected in CSF from 41 of 85 patients (49.4%) and was associated with disease burden and adverse outcome, co-deletion of chromosome arms 1p and 19q, and mutations in the metabolic genes *IDH1* or *IDH2* were shared in all matched ctDNA-positive CSF tumor pairs [98]. In 640 patients with localized tumors such as glioma, the ability to detect tumor alleles (*IDH, p53, EGFR, PTEN*) was limited to a subset of patients [89].

As can be seen in Table 2, numerous resources are used for identification of suitable targets for glioblastoma liquid biopsy. Still, non-specific symptoms of glioblastoma, such as nausea, dizziness, and personality changes, make the early diagnosis of the disease immensely difficult. Current applications of liquid biopsy for glioblastoma diagnosis have to be improved for two main reasons: although in normal physiological conditions the BBB prevents ctDNA circulating into the blood, in pathological situations (i.e., presence of glioblastoma) the BBB is disrupted, which allows for EVs and other cell-free nucleic acids to cross it; and the detection of all cell-free nucleic acids remains a challenge due to the low amounts in the blood in the early course of the disease [2]. However, various new approaches and methods are being developed, such as detecting cell-free nucleic acids (cfDNA and cfRNA) with digital droplet PCR (ddPCR) [99,100] and stabilizing cell-free nucleic acids in the blood [101]. With the development of more sensitive methods, the forthcoming diagnosis of glioblastomas with liquid biopsy will be less demanding. In the future, biomarker-based liquid biopsy systems for glioblastoma diagnosis will accelerate the development of screening programs for early disease detection, which will ultimately result in less deaths.

## 4. Therapy

Current glioblastoma therapy represents a combination of surgery, radiation, and chemotherapy. The latter is based on DNA alkylating agent temozolomide, where overall longer patient survival rates have been shown when epigenetic modification (i.e., methylation of *MGMT* (O^6^-methylguanine–DNA methyltransferase) promoter, leading to reduction in DNA repair) was present at the time of chemotherapy application [102,103]. Another FDA-approved glioblastoma therapy is bevacizumab, which is humanized monoclonal antibody IgG_1_. It is an antiangiogenic therapy blocking tumor vasculature through neutralization of the overexpression of the vascular endothelial growth factor (VEGF) A [104]. Since these treatments are not effective enough, they cause non-specific toxicity and have insufficient pharmacokinetics; therefore, there is a constant search for new treatments with improved efficiency and less adverse effects [105]. Nanomedicine seems to be a promising alternative field, with opportunities in diagnostic and therapeutic approaches. Nanomaterials loaded with drugs could importantly enhance therapy through prolongation of circulation times, more targeted delivery, and better controlled release [106].

### 4.1. Nanotechnology Based Brain Delivery Systems

Nanomaterials are defined as natural, incidental, or manufactured materials composed of particles whose size should be between 1 nm and 100 nm [107]. In biomedical applications, the ideal size is between the 10 and 100 nm in order to circumvent renal clearance and liver capture [108]. The most commonly used materials are based on carbon, silica, or metals, and they can have different shapes, such as spheres, tubes, or rods. Important effects on toxicity and biocompatibility are due to the physical-chemical properties of nanomaterials, such as their size, shape, surface area, chemistry and charge, functional groups, and concentration [109].

When designing a nanovehicle loaded with drugs, several factors have to be taken into account. In general, such nanovehicles possess the characteristics of the nanomaterial itself, rather than the properties of the loaded drug [106]. Important features of nanomaterials are high biocompatibility and low toxicity, and for glioblastoma treatment the ability to cross the BBB [103,109]. BBB is comprised of many different cell types adjacent to tight junctions that represent an extremely restrictive barrier. Only molecules that are smaller than 500 Da and are lipophilic can cross the BBB [109]. In glioblastomas, the nanomedicines have to be able to cross not just the BBB, but also the blood brain tumor barrier (BBTB), which has a more heterogeneous vasculature and different permeability [103]. Namely, blood vessels of solid tumors have abnormal structures and can also produce higher amounts of different vascular permeability factors, which lead to increased vascular permeability. The extravasation and retention of macromolecular drugs in the tumor tissue is known as the enhanced permeability and retention (EPR) effect [110]. Due to leaky endothelium, macromolecules up to 400 nm can cross the barrier [111], while nanomedcines of 5–200 nm supposedly show better EPR effect [112]. Due to the EPR effect, the intravenously administered nanomedicines can passively enter the tumor tissue. Since passive transport of the drugs depends on diffusion only, it is hard to control it. Additionally, some tumors do not show the EPR effect, so the diffusion of a drug into the tumor is ineffective. In order to improve the delivery of the nanomedicines, active targeting strategies have to be applied. In this case, the nanovehicles bear affinity molecules, such antibodies, peptides, or aptamers (DNA). They facilitate more efficient transport through the BBB and deeper tumor infiltration of the drugs, as the nanovehicles bind to the target cell’s receptors and increase the cellular uptake of the drugs [111].

After successful design of a nanomedicine and its systemic administration, it reaches the target cells through passive or active targeting. In the next step, the drug has to be released from the nanoparticle. Precise release of appropriate amount of drug into the tumor is an important challenge of nanomedicine [103]. This can be achieved through different stimuli-regulated nanoparticles, which can be activated through changes of pH, redox state, or enzymes in the tumor micro-environment, or can be triggered through various external stimuli, such as the magnetic field, heat, or light [113]. Several different nanomaterials have so far been considered for glioblastoma treatment, such as liposomes, polymers, and inorganic nanoparticles, and early phase clinical trials are on their way. Different strategies have been applied in glioma treatment with nanoparticles, and it is, therefore, impossible to cover the whole spectrum, however some interesting approaches are represented in the subsections below.

#### 4.1.1. Liposomes

Liposomes are spherical vesicles that mimic the cell membrane, as they are composed of a lipid bilayer of phospholipids [114,115]. They possess several advantages, such as low immunogenicity, good biocompatibility, cell specificity, and drug protection. Many of their physical-chemical characteristics, such as size, charge, and attachment of different ligands, can be tailored in order to produce a nanoparticle for a particular target and improve drug efficacy. Although they are very promising nanovehicles, liposomes have some important shortcomings, such as poor scalability, high cost, short shelf life, and in some cases toxicity and off-target effects [116]. Liposome-based nanomedicines have already been tested in early-phase clinical trials. Currently ongoing trials include single-agent therapy trials of nanoliposomes bearing doxorubicin (NCT02766699) and irinotecan (NCT02022644) [117].

In order to achieve an even more targeted approach, new formulations of liposomes are being constructed. A multifunctional liposomal glioma-targeted drug delivery system of a PEGylated liposome bearing doxorubicin has been modified with two ligands. In order to cross the BBB, p-hydroxybenzoic acid (pHA) with affinity to dopamine receptors was used. The cyclic peptide c(RGDyK) preferentially binds to integrin α_v_β_3,_ which is overexpressed in the BBTB and glioma cells, enabling enhanced paracellular delivery. The in vivo results show that the nanoformulation passed the BBB and BBTB, and accumulated in the glioma. Furthermore, the therapeutic effect on the orthopic glioma model showed that application of nanoformulation increased median survival in comparison to non-functionalized liposomes [118]. In order to prevent development and progression of cancer and chemoresistance, Zhang et al. used the CB5005 peptide [119]. The CB5005 sequence can be divided into two segments, and the peptide itself, therefore, exhibits dual functions in cell membrane penetration and as a nuclear factor kappa light chain enhancer of activated B cell (NF-κB) inhibition, which can lead to suppressed tumor growth and higher sensitivity of tumor cells to chemotherapeutics. The in vivo results showed that modified liposomes accumulated in intracranial glioma and significantly prolonged the survival time of mice. The cytotoxicity assay displayed a five-fold increased efficiency in killing glioma cells compared to non-modified liposomes loaded with doxorubicin. The results of CB5005-modified liposomes bearing doxorubicin may have great potential for cancer treatment. since they have synergistic effects (i.e., they work as a transport system and also as chemotherapy of glioma) [119].

Recently, Jhaveri et al. published two studies on resveratrol, a natural polyphenol with chemopreventive effects in many cancer stages, including initiation, promotion, and progression. They encapsulated resveratrol in liposomes modified with transferrin moieties to enable tumor specific delivery. In contrast to conventional chemotherapeutics that are effective in rapidly dividing cells, resveratrol acts also on tumor-initiating cells (TIC), which are important for the maintenance, invasiveness, and recurrence of glioblastoma. The in vitro study of neurospheres as the TIC model showed that resveratrol inhibited their anchorage-independent growth and they had increased activation of caspases 3/7 [120]. In the in vivo study, significant tumor reduction and increased mean survival time of the animals was shown, supporting the efficacy of resveratrol in glioblastoma [121].

Many other approaches have been tested for liposomes as drug delivery systems, including dual targeting systems of liposomes modified with transferrin for receptor targeting, and increasing translocation with cell penetrating peptide PFVYLI (PFV) for co-delivery of two chemotherapeutics, doxorubicin and erlotinib. Administration of two therapy drugs that act through multiple non-overlapping and synergistic mechanisms is expected to improve therapeutic efficacy and prevent cancer cell drug resistance. The in vitro results showed significant higher apoptosis and also efficient transport through the model BBB of the in vitro brain tumor model [122]. Using the same drug combination of dual-modified liposome with transferrin for receptor-mediated transcytosis and a cell-penetrating peptide penetratin (Pen) for facilitated internalization of delivery, the carrier was tested in vitro and in vivo. In both cases, excellent antitumor efficiency was achieved, supporting the co-delivery of chemotherapeutics for treating glioma [123].

The next step in liposome design is the integration of a stimuli-controlled drug release system. A pH-sensitive liposome loaded with paclitaxel was prepared by Shi et al. [124]. The lipososme was modified with multifunctional peptide TR, a tandem peptide consisting of cRGD and histidine-rich TH peptide, enabling active targeting of integrin αvβ3 for efficient transcytosis across the BBB, and also active and deep target infiltration into the glioma. The prepared liposome efficiently crossed the BBB and showed lower IC_50_ in comparison to the free drug or the liposome with TH only or cRGD peptide for both glioma cells and brain cancer stem cells. The in vivo experiment showed an increased median survival rate of the animals treated with the modified liposome in comparison to free drug. Also, Babincova et al. designed a thermosensitive magnetoliposomes with superparamagnetic iron oxide nanoparticles and doxorubicin, and tested this in vitro and in vivo. Heating to 43 °C enabled controlled release of encapsulated doxorubicin. The results showed inhibition of tumor growth and complete regression [125].

Although the newly designed drug-delivery systems are based on already established liposomes, before any conclusions are made we have to keep in mind that more information on the safety, immunogenicity, interaction with endogenous proteins, distribution, and pharmacokinetics of the transported drug have to be gathered [126].

#### 4.1.2. Polymeric Nanoparticles

Polymeric nanoparticles are biodegradable particles made of a core polymer matrix, where therapeutic agents are encapsulated or can be conjugated onto the surface. They can be composed of synthetic polymers such as poly-ε-caprolactone, polylactides, polyglycolides, and very widely used poly(lactide-co-glycolides) (PLGA); of natural polysaccharides such as chitosan, hyaluronic acid, or corn starch; or of proteins such as albumin, gelatin, collagen, transferrin, and others. The functionalization of polymeric nanoparticles with targeting agents is versatile and can be also stimuli-responsive. The major disadvantage of the polymeric nanoparticle is the burst release of the loaded molecules, and therefore it lacks long-term release, however this can be overcome with appropriate functionalization [115,127].

Among recent studies, Ganipineni et al. applied an interesting strategy of stimuli-induced release of paclitaxel from superparamagnetic iron oxide (SPIO)-loaded PEGylated PLGA-based nanoparticles (PTX/SPIO-NPs). The ex vivo analysis showed enhanced accumulation of the PTX/SPIO-NPs in the brain glioma of orthopic mice. The efficacy of the magnetic targeting treatment in vivo significantly prolonged the median survival time [128]. Lopez-Bertoni et al. just recently discovered cancer stem cell inhibiting miRNAs. They prepared a polymeric nanoparticle of poly(β-amino ester), which enables high efficacy of intracellular delivery, has low cytotoxicity, enables escape from endosomes, and releases the molecules upon environmental triggers. The prepared nanodelivery system was able to co-deliver two different miRNA mimics, the miR-148a and miR-296-5p, into the same glioma cell. For the in vivo experiment, direct injection of the nanoparticle with miRNA into the brain tumor xenografts was used. The co-delivery inhibited the tumor growth and prolonged animal survival time [129]. Among interesting drugs in oncology are histone deacetylase inhibitors, but their use has been limited due to poor delivery into tumors. Householder et al. developed a quisinostat-loaded poly(d,l-lactide)-b-methoxy poly(ethylene glycol) nanoparticle. The in vivo results on orthopic glioma show efficient restriction of tumor growth and prolongation of animal survival time [130].

#### 4.1.3. Solid Lipid Nanoparticles

Solid lipid nanoparticles (SLN) and nanostructured lipid carriers (NLC) are made from natural-based materials or natural lipids. Due to their natural origin, they are non-toxic in extracellular and intracellular environments after degradation and have low immunogenicity. Their small size is another important advantage, as they are not up-taken from the blood flow by the macrophages. SLN and NLC can be modified with different targeting agents, and they have implicit capability for controlled release of different molecules, including therapeutic agents [115]. The important limitation of the SLN is their limited loading space, and in order to overcome this NLC were developed. NLC are made of a mixture of spatially diverse solid and liquid lipids, which arrange in a less perfect crystalline lattice than SLN, leaving more space for cargo loading [131].

Qu et al. constructed three different polymeric nanoparticle carriers (PLGA, SLN, and NLC) to test which one is better for delivering temozolomide to U87 malignant glioma cells and to mice-bearing malignant glioma. All three nanoparticles enabled sustained release of the drug, but the best anti-tumor activity and the most significant tumor inhibition was shown by NLC, which supports the use of NLC for efficient glioma therapy [132]. Furthermore, Zhang et al. designed a dual-ligand modified NLC with lactoferrin and arginine–glycine aspartic acid (RGD) tripeptide. The lactoferrin enables the transport across the BBB and also the passage via receptor-mediated signaling pathways into glioma, while the RGD specifically binds to αvβ3 overexpressed in the neurovascular endothelial cells. The nanosystem showed high encapsulation efficiency in co-loading of temozolomide and vincristine as combination therapy. The in vitro results showed sustained release and significantly higher cellular uptake of the modified NLC in comparison to non-modified NLC, while the dual drug loading showed a synergistic effect. The in vivo results showed the drug delivery system to be very promising, as the tumor inhibition was the highest when animals were treated with the dual-targeting co-delivery system [133].

#### 4.1.4. Polymeric Micelles

Polymeric micelles are lipid-based structures that are spontaneously assembled in water solutions when the critical concentration of amphiphilic molecules is reached. They enable transport of insoluble drugs in their hydrophobic core, while the hydrophilic shell provides stabilization, enabling prolonged circulation times. Micelles enable controlled and sustained release, which can be triggered by different stimuli [115,116].

The improved targeting of the BBB by the polymeric micelles can be achieved with incorporation of cyclic peptides (cRGD) on the surface of the micelles. The cRGD bind to αvβ3 and αvβ5 integrins, which are overexpressed in the tumor neovasculature and in the tumor cells. Recently, an anthracycline epirubicin has been approved by FDA as an anti-glioblastoma agent. The in vivo results of the application of the cRGD-modified micelle into mice-bearing orthopic glioma showed high levels of the drug delivered into the tumor and suppression of its growth [134]. In order to improve the chemotherapy effect, a co-delivery of two synergistic agents (doxorubicin, a topoisomerase II inhibitor; and curcumin, the NF-κB inhibitor) were encapsulated into a PEG-PE-based polymeric micelle. Additionally, a tumor-targeting molecule, a single chain fragment variable (scFv) of monoclonal antibody against glucose transporter 1 (GLUT-1), was added onto the polymeric micelle. The GLUT-1 receptors are overexpressed in the BBB and glioma cells due to enhanced glucose requirements of the cancer cells. The prepared polymeric micelle showed increased cytotoxicity and activation of the caspase 3/7, and deeper penetration into the 3D model tumor mass of U87 spheroids [135].

#### 4.1.5. Dendrimers

Dendrimers are small, globular, highly branched, and symmetrical polymeric molecules with a well-defined structure. They can be used for bioimaging and drug or gene transport [136]. Drugs encapsulated into the dendrimers show better solubility in water, have longer circulation times, and also cause less negative side effects due to chemotherapy. Additionally, dendrimers can covalently link drugs to the exterior of nanoparticles, thus enabling dual-targeting [137]. Dendrimers used in biological application are made of different polymers, among which poly(amidoamine) (PAMAM) dendrimers are preferred, due to their extensive branching, monodispersion, and controlled molecular mass [136,138]. The PAMAM dendrimers are made of an ethylene diamine initiator core, around which amidoamine repeating units are radially attached [139]. In order to prepare a glioma targeting delivery system able to cross the BTBB, a PEGylated (PAMAM) dendrimer conjugated with glioma homing peptides (Pep-1) (Pep-PEG-PAMAM) was prepared. The Pep-1 is a specific ligand for interleukin-13 receptor α2 (IL-13Rα2), which is overexpressed in established glioma cell lines, enabling mediated endocytosis of glioma cells. The in vitro results and in vivo animal model showed that accumulation of the dendrimer was significantly enhanced, along with the penetration at the tumor site, making the Pep-PEG-PAMAM a promising drug delivery system [140]. Fourth-generation PAMAM dendrimers (G4-DOX-PEG-Tf-TAM) conjugated with transferrin (Tf) have been used as dual-targeting drug carriers. Two drugs were loaded into the system (namely tamoxifen and Tf, which are used for TfR-mediated endocytosis) and bonded to the exterior, while doxorubicin was encapsulated in the core. The in vitro experiment showed efficient crossing of the model BBB and uptake in glioma C6 cells, which confirms that the dual system has the potential to target glioma cells [137]. Dendrimers have been used to transport small interfering RNA (siRNA) that have a great potential for silencing the up-regulated genes or genes involved in cell division in cancer cells. Two different therapeutic siRNA molecules, B-cell lymphoma/leukemia-2 (Bcl-2) siRNA and vascular endothelial growth factor siRNA, were complexed with fifth-generation PAMAM dendrimers, which were modified with β-cyclodextrin (β-CD) and had entrapped gold nanoparticles (Au DENPs). The results showed efficient delivery of the siRNA into the glioma cells and enhanced gene silencing [141].

Short life expectancy and poor quality of life in patients show the importance of development of new methods for glioblastoma therapy, some of which are summarized in Table 3. Despite substantial research, there is still an unmet need for the design of modern therapeutic strategies for glioblastoma. Additionally, the majority of studies are still on the waiting list for clinical testing.

### 4.2. Theranostics

Theranostics describes the use of a single agent for two purposes: diagnosis and therapy. This concept has also been explored in the development of nanoparticles for glioblastoma management. An example of theranostic applications of nanoparticles in glioblastoma management is a nanosystem composed of iron oxide nanoworms coated with two branched chimeric peptide: one branch is the tumor-specific vascular homing element CGKRK (Cys-Gly-Lys-Arg-Lys) and the other is the membrane-perturbing proapoptotic D-amino acid peptide _D_[KLAKLAK]_2_ that serves as a drug [142]. The _d_[KLAKLAK]_2_ is a synthetic antibacterial peptide that causes disruption of the mitochondrial membrane when internalized into eukaryotic cells, and therefore initiates apoptosis [143]. The system was tested on a murine glioblastoma model and the authors reported high tumor specificity leading to reduced general toxicity [143]. The nanoworm-CGKRK-_D_[KLAKLAK]_2_ system showed selectivity for tumor cells and inhibited tumor growth in vivo. Moreover, due to the presence of iron oxide (which can be used as an imaging agent in MRI) and _D_[KLAKLAK]_2_ peptide as the cytotoxic drug, the nanoworm system can potentially be applied for both diagnosis and therapy, respectively.

Photothermal therapy is considered to be a promising tumor therapy because of its minimal invasiveness to adjacent tissue, selectivity for tumor cells, and great efficacy against cancer resistance [144]. In cancer treatment, photothermal therapy relies on localized heating induced by a near-infrared (NIR) beam of light that deeply penetrates tissues. The mechanism of action lies in the induction of apoptosis with activation of the caspase-3 pathway [145]. Among a variety of NIR transducers, gold nanostars (AuNSs) have gained popularity due to their high absorbance and excellent photothermal conversion efficiency, which makes them suitable photothermal agents for disease diagnosis and therapy. For this purpose, Wang et al. developed a nanocomposite of AuNS@ probes with fluorescent “turn on” gold nanostars for photothermal therapy, along with self-theranostic feedback based on “turning on” fluorescence for caspase-3 imaging [144]. The authors evaluated the cyto- and photo-toxicity of AuNSs and AuNS@ probes using U87MG glioblastoma cells. Their results show a concentration-dependent photo-killing effect.

Another method is theranostic near-infrared photoimmunotherapy (NIR-PIT), which is a combination of target-specific antibodies and local application of NIR light [146]. This method has numerous advantages, among which targeting antigens that are not exclusive to tumor cells while leaving neighboring cells unharmed and theranostic applications are the most important advantages. In their study, Jing et al. used NIR-PIT to destroy glioblastoma stem cells with targeting AC133, the stem-cell-specific epitope of CD133 [146]. By using PIT, the authors successfully eradicated patient-derived stem cells and showed strong anti-tumor effects in established aggressive brain tumors. In addition, the authors suggest the use of NIR imaging for fluorescence guided tumor resection and pathological determination of tumor margins in humans in the future.

Photodynamic therapy refers to the selective uptake of photosensitizing molecules in a tumor activated by light of an appropriate wavelength [147]. Upon activation, the photosensitizer interacts with molecular oxygen and leads to the formation of cytotoxic short-lived singlet oxygen. This causes apoptotic and necrotic responses in the tumor. Even though photofrin is the most widely used photosensitizer, it has its limitations, including prolonged skin photosensitization, meaning patients should avoid direct exposure to sunlight for 4 to 6 weeks after treatment. To this end, nanoparticle platforms composed of polyacrylamide that can be loaded with photoactivatable agents for delivery of singlet oxygen have been explored [148].

## 5. Immunotherapy in Glioblastoma

The brain was traditionally known as an immune-privileged organ [149]. Namely, the BBB limits the access of immune cells, there is no lymphatic drainage, antigen-presenting cells are rare, and there is downregulation of the major histocompatibility complex (MHC) [149]. However, some of the brain pathologies, such as multiple sclerosis and neurodegenerative diseases, have been known for their active immune systems. Besides, a disrupted BBB, such as through injury, enables immune system cells to enter the brain [150]. Today, it is known that the brain has both innate and adaptive immune systems [149].

Glioblastomas are characterized by their highly immunosuppressive nature, which is a result of downregulation of MHC, increased expression of death ligand-1, and increased recruitment of regulatory T cells [151]. Cells in the glioblastoma microenvironment release cytokines such as TGFβ, interleukin-10 (IL-10) and VEGF, which have immunosuppressive roles [150]. The activity of monocytes and dendritic cells is usually decreased and the number of infiltrating T cells is reduced [150,152]. Moreover, one of the most represented cells in glioblastomas are tumor-associated macrophages that are either immunopermissive (M1 type) or immunosuppressive (M2 type). The presence of immunosuppressive tumor-associated macrophages is linked to poorer patient outcome [151,153]. Also, M1 prevalence in tumors is related to anti-tumor properties and potential reduction of tumors [154]. Systemically, glioblastoma patients have abnormal cellular immune systems. Patients have increased expression of programmed cell death 1 ligand (PD-L1) in macrophages in peripheral blood [155]. Additionally, T cells are sequestered from blood into the bone marrow due to downregulation of sphingosine 1-phosphate receptor 1 (S1PR1) [152]. Glioblastoma patients are usually treated by chemotherapy and radiotherapy together with glucocorticoids, and they can potentially change immune system status [149]. Radiotherapy and chemotherapy have been shown to increase PD-L1 expression and activate immunosuppressive macrophages [156,157]. The distinctive immunosuppressive nature of glioblastoma has stimulated the development of various immunotherapeutic approaches, which are explained in the following section and are also schematically presented in Figure 2.

### 5.1. Chimeric Antigen Receptor T-cells

Chimeric antigen receptors (CAR) are synthetically derived receptors that recognize specific surface proteins. They are formed in the extracellular domain, consisting of a signalling peptide, antigen-recognition part, and a spacer. The signal peptide directs protein into endoplasmic reticulum and the antigen recognition region is usually formed from scFv. The transmembrane region is related to the stability of the receptor, and currently the CD28 transmembrane region presents as the most stable. The intracellular part consists of CD3 ζ, which includes immunoreceptor tyrosine-based activation motifs [158]. The intracellular domain in third-generation CAR has two costimulatory domains as well as CD3ζ, such as CD3ζ-CD28-OX40 or CD3ζ-CD28-41BB [158]. In practice, patients’ T-cells are collected, enriched, and specific T-cell subtypes are selected. T-cells are then stimulated and genetically engineered, for instance by lentiviral or retroviral transduction. Afterwards, engineered T-cells are amplified in a special medium that supports cell-growth. Lastly, the T-cells are injected back into the patient [159]. In glioblastoma, CAR-T against antigen IL13Rα2, human epidermal growth factor receptor 2 (HER2), and EGFRvIII have been developed [160].

IL13Rα2 is a high affinity IL-13 receptor that is overexpressed in glioblastoma, including glioblastoma stem cells, and it is related to poor survival [159]. Its expression is associated with the mesenchymal subtype of glioblastomas, which is also characterized by its proinflammatory nature. Moreover, its expression is low in normal brain tissue [161]. A few clinical trials have been reported applying IL13Rα2 CAR-T in glioblastoma treatment (Table 4). Brown et al. developed a IL13Rα2 CAR-T complex, named IL13-zetakine, which has reduced binding potential to the widely expressed IL13Rα1/R4α complex [161]. Three recurrent glioblastoma patients were enrolled in the phase l study. Patients underwent collection of peripheral mononuclear cells, which were used in vitro to engineer CD8+ cytotoxic T lymphocytes expressing IL13-zetakine. A Rickham catheter was inserted during surgery. When patients recovered, T-lymphocytes were injected into the catheter. The patients did not have any serious side effects and showed a therapeutic response [161]. In another clinical trial (NCT02208362) that is still ongoing, a case report has been published. A recurrent glioblastoma patient received intracranial infusion of IL13Rα2 CAR-T followed by infusions in the ventricular system. After the therapy, the patient did not have any serious side effects and tumors were undetectable by PET. This was a case of a complete response. Unfortunately, after 228 days the disease recurred. Follow-up analyses suggest that tumors in new locations had decreased expression of IL13Rα2 [162].

Application of CAR-T therapy in glioblastomas raises several questions and concerns that need to be answered. First of all, glioblastomas have a highly immunosuppressive nature that seems to be preserved after CAR-T therapy [163]. Moreover, glioblastomas are an exceptionally heterogeneous disease both inside the tumor and between patients, and also after the expression of IL-13 Rα2 and EGFRvIII, which have previously been targeted in CAR-T therapy [163].

#### Nanoparticles Improve T-cell Therapy

When T-cells are being processed for application, they need to be expanded ex vivo. To improve expansion and lower the cost, various nanoparticles that act as antigen-presenting parts have been proposed, examples of which are given in Table 5. These include polystyrene beads, liposomes, iron/dextran nanoparticles, and carbon nanotubes. The most attention has been focused on Dynabeads, commonly composed of Fe_3_O_4_ in polystyrene matrix, with anti-CD3 and anti-CD28. They have shown consistent results for T-cell expansion, handling them is easy, and the costs are lower [164]. Similarly, Fadel et al. developed carbon nanotubes with attached antigens. Moreover, IL-2 was packed in poly(lactide-co-glycolide) nanoparticles together with magnetite. The authors showed that the nanoparticles successfully promoted T-cell expansion with 1000-fold lower usage of IL-2 [165]. Nanoparticles are also a potential tool for studying and tracking T-cell in vivo. Meir et al. engineered T-cells that expressed melanoma-specific T-cell receptors and labelled them with gold nanoparticles. The T-cells were then injected into mice bearing melanoma, and CT enabled tracking of T-cells. The authors showed that engineered cells accumulated at the tumor site, in contrast to non-engineered cells. The importance of the developed method is the ability to study the kinetics and biodistribution of T-cells [166].

Nanoparticles can also be administered together with the T-cells into the body. Tsao et al. developed a degradable hydrogel composed of poly(ethyleneglycol)-*g*-chitosan. The gel exists in liquid state at low temperatures and at higher temperatures it forms a gel. The purpose of the gel was to encapsulate T-cells to control their release for glioblastoma treatment. The authors showed that lymphocytes released from the gel successfully killed glioblastoma cells in vitro, and the use of poly(ethyleneglycol)-*g*-chitosan was more efficient than in Matrigel. The advantages of poly(ethyleneglycol)-*g*-chitosan are its biocompatibility, degradability, and low immunogenicity [167]. Another mechanism to improve T-cell therapy is its application in photothermal therapy. Chen et al. developed poly(lactic-*co*-glycolic) acid nanoparticles loaded with indocyanine green, a NIR dye. The nanoparticles were injected into mice with melanoma tumors. Subsequently, CAR-T against chondroitin sulphate proteoglycan-4 was also injected. The efficacy of T-cells was increased after photothermal therapy, with the combined therapy reaching a therapeutic effect [168].

### 5.2. Immune Checkpoint Modulators

Tumors usually develop several mechanisms to avoid elimination by immune system. One of the mechanisms involves inhibition of T-cell response. T-cells are activated when specific antigens are presented to them, and in turn they destroy tumor cells. However, the presence of co-inhibitory molecules, such as cytotoxic T-lymphocyte-associated antigen 4 (CTLA4) and PD-1, prevent cytotoxic T-cell response [176]. Therefore, molecules targeting CTLA4 and PD-1 have huge potential in cancer therapy. A monoclonal antibody drug targeting CTLA4, ipilimumab, has already been approved by the FDA and European Medicines Agency (EMA) for melanoma treatment. Another CTLA4 antibody, tremelimumab, has been approved for mesothelioma treatment [177]. Also, anti-PD1 antibodies nivolumab and pembrolizumab have been approved by the FDA in melanoma or non-small-cell lung carcinoma (NSCLC) treatment [176]. Anti-PD-L1 antibodies atezolizumab and avelumab have also been approved [177]. Indeed, there are numerous pre-clinical experiments and clinical trials evaluating different immune-checkpoint modulators in cancer treatment.

PDL1 is highly expressed in glioblastomas. Namely, expression of PDL1 is expressed in 88% of samples of newly diagnosed glioblastomas and in 72% of samples of recurrent glioblastomas. The expression is also very low in surrounding healthy tissue. Moreover, the expression is also significantly greater than in melanoma and NSCLC (approximately 30% and 25%–36% greater, respectively). In glioma cells, PDL1 inhibits activation of T-cells and production of IFN-γ, IL-2, and IL-10 by lymphocytes [176]. Due to high expression of PDL1 in glioblastoma, much effort has been focused on evaluating anti-PD1/PDL1 monotherapy or combined therapy [176].

In glioblastoma, it seems that immune checkpoint modulator monotherapy will not be sufficient. Namely, nivolumab and pembrolizumab monotherapy in recurrent glioblastoma patients did not show any benefit in terms of patient survival [178]. There are many ongoing clinical trials, such as for immune checkpoint modulators combined with chemotherapy, radiotherapy, bevacizumab, or other therapeutic modalities [178]. In addition, many other potential modulators, such as those targeting indoleamine 2,3-dioxygenase (IDO), have been developed. They are being evaluated in clinical trials with patients with solid tumors or glioblastomas [178]. In general, until now, immune checkpoint modulators have not proven to be effective in glioblastoma treatment. They have, however, proved to be safe and tolerable for clinical application [178].

#### 5.2.1. Nanoparticles and Immune Checkpoint Modulators

The efficacy of immune-checkpoint inhibitors in glioblastoma treatment may be improved by using antigen-capturing nanoparticles. These are nanoparticles that have modified surface and are able to bind tumor antigens. The surface is usually changed by non-covalent hydrophobic–hydrophobic interactions, or ionic or covalent interactions [179]. Min et al. developed a nanoparticle formed from poly(lactic-co-glycolic) acid. They showed that different surfaces have different abilities in antigen capturing. Further, 1,2-dioleoyloxy-3-(trimethylammonium)propane and poly (lactic-co-glycolic) acid had the greatest ability and successfully bound the tumor antigens. The nanoparticles were then injected in mice with melanoma that underwent anti-PD-1 therapy. The nanoparticles successfully presented the antigens to dendritic cells, as the complete response rate was 20%. Overall, the study showed that biodegradable and biocompatible nanoparticles can enhance presentation of antigens and significantly improve survival [169]. Zhang et al. developed cell membrane-derived vesicles that expressed the PD-1 receptor and were packed with indoleamine 2,3-dioxygenase (IDO) inhibitor. Vesicles were then injected into mice bearing melanoma tumors. Combined therapy was more efficient than single therapy, and the therapy significantly reduced tumor growth and improved the survival of mice [170].

#### 5.2.2. Immune Photothermal Therapy

The purpose of immune photothermal therapy is to eradicate both primary tumors and tumors at different locations. Liu et al. developed a combined therapy using immune checkpoint inhibitors in combination with plasmonic gold nanostars, which are applied in photothermal therapy [171]. Gold nanostars are potent in converting light energy into heat. The therapy was tested in bladder cancer in vivo and showed very promising results. In general, the combined therapy showed better results than anti-PD-L1 monotherapy. The survival rate was 40% and the mice developed long-lasting immunity against cancer cells [171]. In another study, Peng et al. developed NLG919/IR780 micelles of less than 50 nm in size. IR780 can absorb light in near-infrared regions and NLG919 is an IDO inhibitor. Micelles were applied to mice bearing breast tumors. The combined photothermal therapy and immunotherapy successfully killed primary tumors and prevented the growth of secondary tumors. Moreover, the activity of T-reg was decreased and presence of CD8^+^ T-cells was increased [172]. In a study by Luo et al., hollow gold nanoshells were packed together with anti-PD1 peptide in poly(lactic-co-glycolic acid) nanoparticle. Gold nanoshells were used in photothermal therapy and anti-PD-1 was used in immunotherapy. Then, the nanoparticles were injected into mice bearing tumors. The mice were radiated with near-infrared light, which also enabled the release of peptides from the nanoparticles. Later, CpG was added to poly(lactic-co-glycolic acid), which served as a vaccine, and the combined therapy induced maturation of dendritic cells and release of cytokines. The results were very promising because the therapy diminished primary tumor, inhibited growth of metastases, and also prevented formation of new tumors [173].

### 5.3. Vaccine-Based Immunotherapy

Vaccine-based immunotherapy has mostly been studied in rodent animal models and has included several strategies, such as modifying glioma cells, dendritic cell application, peptide-based vaccines, or combined approach with other treatment modalities [180]. In glioblastomas, the most common targets include mutant IDH, EGFRvIII, a panel of antigens, or even personally selected antigens. In glioblastoma, the *EGFR* gene is amplified in 40% of cases, and more than 50% of cases include deletion or mutation of exon 2-7 [180,181]. The mutated form of the protein does not have a ligand-binding domain and results in constitutive activity, which in turn promotes malignancy. Also, the mutated receptor can be activated by several kinases, such as Src family kinases [181]. The changed amino-acid sequence has been discovered to be immunogenic, and a vaccine called rindopepimut, based on the mutated peptide sequence, was developed to induce immune system [180]. The initial clinical trials showed benefits from the vaccine. However, the subsequent phase III clinical trial failed to show benefits to overall survival [182]. In another clinical trial, named ReACT (A Study of Rindopepimut/GM-CSF in Patients With Relapsed EGFRvIII-Positive Glioblastoma), patients with recurrent glioblastomas received either control or rindopepimut. The vaccine showed benefits over control, as median survival was 12 months versus 8.8 months, respectively [183].

#### Nanoparticles Increase Vaccine Efficacy

The efficacy of vaccines can be increased by application of nanoparticles. They can especially protect the vaccine from degradation and enhance APC uptake. Kuai et al. developed nanodiscs based on lipids and peptides derived from high-density lipoprotein. Then, antigen peptides and cholesterol were bound to the surface. In mice bearing melanoma tumors, the nanodiscs showed higher efficacy in T-cell priming and the effect lasted longer. In mice bearing tumor models, nanodiscs together with anti-PD-1 therapy caused tumor regression in 88% of mice, which was significantly higher than in either of them separately [174]. The vaccine can also be in the form of mRNA. Liu et al. formed lipid/calcium/phosphate nanoparticles that delivered mRNA-encoding MUC1. The vaccine was then injected into mice bearing triple-negative breast cancer together with anti-CTLA-4 antibody. The combined therapy showed better response than either therapies separately [175].

## 6. Conclusions

Glioblastoma disease is the most aggressive brain tumor, and current therapy is rather ineffective, meaning patient survival rates and prognosis are poor. A way to overcome these issues is the development of nanomedicines, which would improve diagnosis, allow for efficient drug targeting and transport through the BBB, improve drug solubility, prolong blood circulation half-life, and enable controlled and sustained drug release Many different formulations of nanomedicines are being tested on cell lines and animal models in order to find a combination of nanomedicine that has low toxicity, is biocompatible, and is specifically directed towards tumor cells. The only nanomedicines that are currently in clinical trials are PEGylated liposomes with doxorubicin, both as a single-agent therapy and in combination with temozolomide. Introduction of fluorescent probes as imaging agents for intraoperative MRI shows promise for increasing quality of life, resulting from maximizing the extent of the surgical resection. Furthermore, personalization of these probes is possible, which will lead to tailoring the diagnostic and treatment processes to the patients’ needs.

Use of less invasive approaches such liquid biopsy and subsequent analysis of cfDNA to confirm diagnosis of present mutations and monitor tumor evolution and response to therapy would significantly improve the clinical management of glioma patients. The development of immunotherapeutic approaches in cancer treatment, including glioblastoma, has expanded in recent years. However, treatments have not shown much success yet. The main reasons for the failure are their highly immunosuppressive nature and presence of the BBB. One of the proposals to improve immunotherapy in glioblastomas is application of nanoparticles. They especially showed benefits in photothermal therapy in combination with immune checkpoint inhibitors.

In general, designing effective treatments for glioblastoma disease is challenging due to the high tumor heterogeneity, meaning that among different cells present in the tumor, the chosen therapy could be effective for one cell type but ineffective for others. This is why multiple combined approaches are also taken into consideration and are believed to be more successful in eradicating diseased cells. Smart application of nanoparticles in precise medicine will eventually lead to higher percentages of long term survivors.

## Figures and Tables

**Figure 1 molecules-25-00490-f001:**
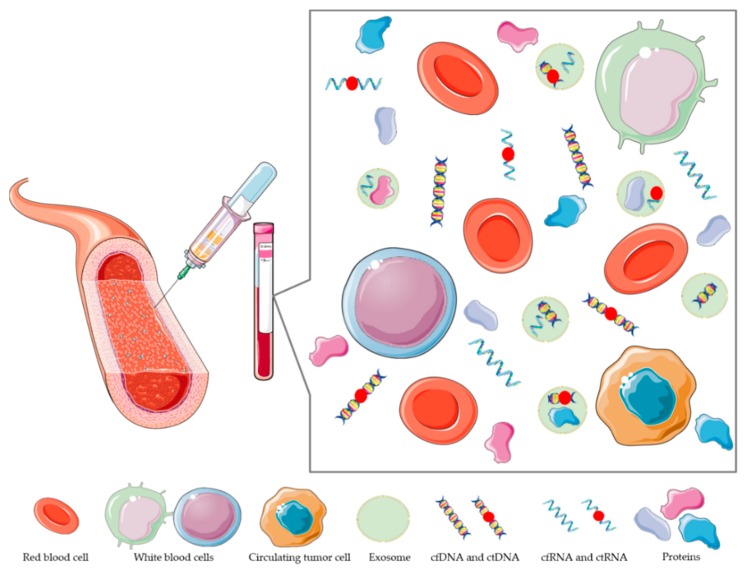
Schematic presentation of blood content. To obtain tumor information circulating tumor cells, circulating tumor DNA (ctDNA) and exosomes are analyzed from blood biopsy. The picture is for graphical illustration only, and does not represent actual sizes, size ratios among particles, or quantity of analytes. The image was created using Servier Medical Art (SMART) (https://smart.servier.com/). Servier Medical Art by Servier is licensed under a Creative Commons Attribution 3.0 unsupported license.

**Figure 2 molecules-25-00490-f002:**
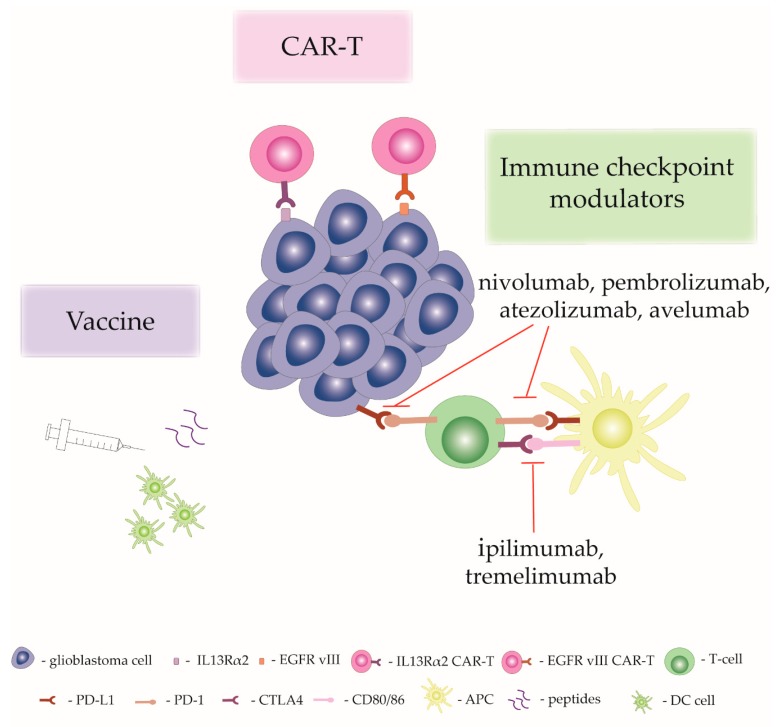
Presentation of the most common immunotherapeutic approaches in glioblastoma treatment. Note: CAR-T = immune checkpoint modulators and vaccine; CAR = chimeric antigen receptor; EGFR vIII = epidermal growth factor receptor vIII; IL13Rα2 = interleukin-13 receptor alpha-2; CTLA4 = T-lymphocyte-associated antigen 4; APC = antigen presenting cell; DC = dendritic cell; PD-1 = programmed cell death protein 1; PD-L1 = programmed cell death 1 ligand.

**Table 1 molecules-25-00490-t001:** Nanoparticles currently tested for glioblastoma imaging.

Nanoparticle	System	Properties	Reference
EGFRvIII antibody—IONP complex	In vitro In vivo (murine models)	Decreases glioblastoma cell survival No significant toxicity on astrocytes or murine models	[18]
IONPs with infrared core functionalized with sd antibody targeted against IGFBP7	In vivo (murine models)	Enhances visualization in preoperative or intraoperative MRI Significant retention of IONP-sd α-IGFBP7 Ab in tumor compared to non-targeted IONP Increases the extent of surgical resection	[11]
Ultrasmall superparamagnetic iron-oxide-based nanoparticles	Human subjects	Visualization of lesions with damaged BBB Does not leak out of the blood vessels early after injection Delayed and prolonged enhancement Suitable for dynamic studies, such as perfusion imaging	[19]
5-ALA for FGS	Human subjects	Allows for distinguishing glioblastoma from normal brain during surgery Improves complete resection rate Leads to longer overall survival	[20]
5-ALA	Human subjects	Able to detect infiltrating glioma cells at tumor margins Positively associated with increased cell proliferation	[21]
LED headlamp for 5-ALA-guided glioblastoma resection	Human subjects	Greater freedom of movement Inexpensive Complementary to microscopy-based surgery Suitable for use in resource-restricted areas	[22]
SERRS-MSOT-nanostars	In vivo (murine models)	Sensitivity in detecting diffuse glioblastoma margins Selective accumulation in glioblastoma and glioblastoma-periphery	[23]
Raman active SERRS nanotags targeted against integrin receptors	In vivo (murine models)	Non-invasive visualization of glioblastoma High contrast and signal specificity for tumor area	[24]
NCs and ANCs coated with dOA or BSA	In vitro In vivo (murine models)	Selectivity towards malignant mass Suitable for visualization and treatment of neoplastic masses	[25]
MSN-labeled neural stem cells	In vitro In vivo (murine models)	Minimally toxic for NSCs Stable in vitro and in vivo Feasible for NSCs-mediated therapy Suitable for drug-loading	[26]

Note: EGFRvIII = epidermal growth factor receptor variant III; IONPs = iron-oxide nanoparticles; sd = single domain; IGFBP7 = insulin-like growth factor binding protein 7; MRI = magnetic resonance imaging; Ab = antibody; BBB = blood–brain barrier; 5-ALA = 5-aminolevulinic acid; FGS = fluorescence-guided surgery; LED = light emitting diode; SERRS-MSOT = surface-enhanced resonance Raman spectroscopy combined with multispectral optoacoustic tomography; NCs = nanocubes; ANCs = assembled larger nanocube constructs; dOA = double oleic acid; BSA = bovine serum albumin; MSN = mesoporous silica nanoparticle; NSCs = neural stem cells.

**Table 2 molecules-25-00490-t002:** Potential targets in glioblastomas for diagnosis with liquid biopsy.

Analyte in LB	LB System	Study	Reference
EGFRvIII, GFAP	In vitro glioblastoma EV	First comprehensive profiling of EV	[51]
*EGFRvIII*	serum EV	Robust method incorporating two different primers	[55]
*wtEGFR*	CSF EV	Significantly expressed *wtEGFR* in CSF EVs	[57]
SDC-1	Plasma EV	Discriminating between high-grade and low-grade glioblastoma Plasma EV SDC1 level decreased after surgery	[59]
Mut-*IDH*	CSF EV	Quantity of mutant *IDH1* transcripts directly correlates with the tumor volume	[63]
miRNA of wt-*IDH*, mut-*IDH*	Serum EVs	Distinguishing preoperative glioblastoma patients from healthy controls	[65]
miR-21	CSF EV	Distinguishing CSF EVs derived from glioblastoma patients and non-oncologic patients Follow-up of miR-21 levels after surgery	[52]
miR-21	Serum EV	Significantly upregulated	[66]
miR320, miR547-3p, RNU6-1	Serum EV	EVs associated with glioblastoma diagnosis and overall survival	[67]
miR-301	Serum EV	Elevated in glioma patients, predictive of overall survival	[68]
Nestin, Human telomerase markers	Blood-CTC	Detecting CTC	[71]
iChip	Blood-CTC	Negative approach isolation—removal of small parts in blood and white blood cells	[7]
EGFR amp	Blood-CTC	Detected in different grades Release of CTCs was associated with EGFR gene amplification	[74]
cfDNA	Blood	Good tendency between cfDNA of glioma treatment course and response	[88]
ctDNA	Plasma	Detection of tumor alleles in 640 patients with different localized tumors (*IDH, p53, EGFR, PTEN*)	[89]
ctDNA	Serum	Detection of ctDNA in 55% of glioma patients	[90]
ctDNA	Plasma	Detection of alterations in 27% of glioblastoma patients *p53* and *NOTCH1* anomalies	[91]
ctDNA	Blood	51% ctDNA/cfDNA detection in glioblastoma patients	[92]
ctDNA	Blood	50% of patients had 1 or >1 somatic alteration detected, highest in meningioma (59%) and glioblastoma (50%), single nucleotide variants were detected in 61 genes	[93]
ctDNA	CSF	Simultaneous testing of seven genes (*IDH1*, *IDH2*, *TP53*, *ATRX*, *TERT*, *H3F3A*, *HIST1H3B*) in diffuse gliomas	[97]
ctDNA	CSF	Detected in 49.4% glioma patients: codeletion 1p19q, mut-*IDH1*, or mut-*IDH2*	[98]

Note: LB = liquid biopsy; EGFRvIII = epidermal growth factor receptor variant III; GFAP = glial fibrillary acidic protein; EV = extracellular vesicle; wt = wild type; CSF = cerebrospinal fluid; SDC1 = Syndecan 1; IDH = isocitrate dehydrogenase; mut = mutant; RNU6-1 = RNA, U6 Small Nuclear 1; CTC = circulating tumor cell; iChip = isolation chip; cfDNA = cell-free DNA; ctDNA = cell-free tumor DNA; p53 = tumor protein p53; PTEN = phosphatase and tensin homolog; NOTCH1 = Notch homolog 1; ATRX = ATP-dependent helicase; TERT = telomerase reverse transcriptase; H3F3A = histone H3.3; HIST1H3B = histone H3.1.

**Table 3 molecules-25-00490-t003:** Selected studies of nanoparticles currently being tested for therapy.

Nanoparticle	Stage	Advantages	Reference/Clinical Trial Identifier Number
Cerebral EnGeneIC delivery vehicle (EDV) (EGFR(V)-EDV-Dox)	Clinical trial Phase I	Single-agent therapy. Intravenous.	NCT02766699
Nanoliposomal irinotecan	Clinical trial Phase I	Single-agent therapy. Intracranial.	NCT02022644
DOX-loaded PEGylated liposomes modified with p-hydroxybenzoic acid (pHA) and c(RGDyK)	In vitro (glioblastoma cells (U87), brain capillary endothelial cells (bEnd.3), and umbilical vein endothelial cells (HUVECs)) In vivo (orthopic mouse model)	Efficient targeting of the tested cell lines and increased doxorubicin cytotoxicity. Passing the BBB and BBTB in vitro and in vivo. Increased median survival.	[118]
DOX-loaded PEGylated liposomes conjugated with CB5005 peptide	In vitro (glioma cells U87) In vivo (xenograft- and intracranial-glioblastoma-bearing nude mice)	Penetrated into glioma cells and delivered DOX into the nucleus. Increased the efficiency of killing glioma cells. In vivo distributed into the brain and accumulated at tumor xenograft. Prolonged the survival time.	[119]
Resveratrol-loaded PEGylated liposome targeted with transferrin	In vitro (glioblastoma cells (U87) and neurospheres) In vivo (xenograft mouse model)	Good drug-loading capacity and prolonged drug-release. Significantly more cytotoxic and induced higher levels of apoptosis compared to free RES. Inhibited tumor growth and prolonged survival time.	[120]
DOX- and erlotinib-loaded PEGylated liposomes, modified with transferrin and cell-penetrating peptide PFVYLI	In vitro (glioblastoma cells (U87), brain capillary endothelial cells (bEnd.3) and in vitro brain tumor model In vivo (orthopic mouse model)	Efficient internalization of drugs and higher apoptosis. Translocation across the BBB. Increased drug accumulation in mice brain and in median survival time.	[122,123]
PTX- and SPIO-loaded, PEGylated, PLGA-based nanoparticles	In vitro (glioblastoma cells (U87)) In vivo (orthopic mouse model)	Enhanced accumulation of nanoparticles in the brain. Prolonged the median survival time. No induced systemic toxicity.	[128]
Temozolomide loaded PNPs, SLNs, and NLCs	In vitro (glioblastoma cells (U87)) In vivo (malignant glioma-bearing mice)	NLCs showed most efficient delivery of temozolomide and higher inhibition efficacy of tumor growth.	[132]
Epirubicin-loaded polymeric micelles decorated with cRGD	In vitro (glioblastoma cells (U87)) In vivo (orthopic mouse model)	Faster and higher penetration of cRGD-decorated NPs into the cells compared to non-decorated NPs. Effective suppression of the tumor growth.	[134]
DOX- and curcumin-loaded polymeric micelles decorated with GLUT1	In vitro (glioblastoma cells (U87))	Successful apoptosis enhancement due to combinatory treatment. Deeper penetration into the 3D spheroid model.	[135]
DOX- and tamoxifen-loaded PAMAM dendrimer conjugated with transferrin	In vitro (glioblastoma cells (U87))	Effective transport across the BBB.	[137]

Note: EGFR = epidermal growth factor receptor; BBB = blood brain barrier; BBTB = blood brain tumor barrier; c(RGDyK) = cyclic peptide composed of RGDyK; DOX = doxorubicin; RES = resveratrol; PTX = paclitaxel; SPIO = superparamagnetic iron oxide; PLGA = poly(lactic-co-glycolic acid); PNPs = polymeric nanoparticles; SLNs = solid lipid nanoparticles; NLCs = nanostructured lipid carriers; cRGD = cyclic peptide composed of RGD; NP = nanoparticle; GLUT1 = glucose transporter-1; PAMAM = PEGylated polyamidoamine.

**Table 4 molecules-25-00490-t004:** Immunotherapeutic approaches in glioblastoma therapies that have entered clinical trials. Not all of the existing clinical trials are presented. Clinical trial numbers were accessed from https://clinicaltrials.gov/.

Immunotherapy Approach	Target	Clinical Trial Number
CAR-T	IL13Rα2	NCT04003649 NCT02208362
CAR-T	HER2	NCT01109095
CAR-T	EGFRvIII	NCT02664363 NCT02844062 NCT02209376 NCT03283631
Ipilimumab	CTLA-4	NCT03430791 NCT03233152 NCT03460782 NCT02017717 NCT04145115
Tremelimumab	CTLA-4	NCT02794883
Nivolumab	PD-1	NCT02550249 NCT02529072 NCT02335918
Pembrolizumab	PD-1	NCT03311542 NCT02337491 NCT03899857 NCT04118036 NCT03661723
Atezolizumab	PD-L1	NCT03158389 NCT03673787 NCT03174197
Avelumab	PD-L1	NCT02968940 NCT03750071 NCT03047473
Rindopepimut (mutated EGFR vaccine)	X	NCT01498328 NCT01480479 NCT00458601

Note: CAR = chimeric antigen receptor; IL13Rα2 = interleukin-13 receptor alpha-2; HER2 = human epidermal growth factor receptor 2; EGFR vIII = epidermal growth factor receptor vIII; CTLA4 = T-lymphocyte-associated antigen 4; PD-1 = programmed cell death protein 1; PD-L1 = programmed cell death 1 ligand.

**Table 5 molecules-25-00490-t005:** Potential use of nanoparticles in immunotherapy.

Nanoparticle	Purpose	Reference
Dynabeads with anti-CD3 and anti-CD28	T-cell expansion	[164]
IL-2 packed in poly(lactide-co-glycolide) nanoparticles together with magnetite	T-cell expansion	[165]
T-cell receptor labelled with gold	T-cell tracking	[166]
Hydrogel composed of poly(ethyleneglycol)-*g*-chitosan	Control release of T-cell	[167]
Poly(lactic-*co*-glycolic) acid nanoparticles loaded with indocyanine green	Photothermal therapy	[168]
NP formed of poly (lactic-co-glycolic) acid	Antigen binding	[169]
Cell-membrane derived vesicles	Therapy	[170]
Gold nanostars	Photothermal therapy	[171]
NLG919/IR780 micelles	Photothermal therapy	[172]
Gold nanoshells	Photothermal therapy	[173]
Nanodiscs	Presentation of antigens	[174]
Lipid/calcium/phosphate nanoparticle	Delivery of mRNA-encoding MUC1	[175]

Note: IL-2 = interleukin 2; NP = nanoparticle; MUC1 = mucin 1, cell surface associated.

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
