# Peer review of "Nanomedicine and Immunotherapy: A Step Further towards Precision Medicine for Glioblastoma"

_molecules, 2020, doi:10.3390/molecules25030490_

Round 1

Reviewer 1 Report

This is a rather long review summarizing the role of nanoparticles and immunotherapy in the treatment of glioblastoma multiforme (GBM). The relevance of section 3 (Liquid biopsies) is not entirely clear since, as the authors state in the last sentence of the section, application of this technique for GBM is a difficult task. The information presented suggests that this approach has not been very successful so far. Therefore I recommend shortening this section to a few paragraphs summarizing the utility of liquid biopsy for GBM.

2, line 56: The authors state that “the majority of glioblastoma patients succumb to the disease within 5 years.” Although true, it’s customary to state the median survival. Please state median survival based on the most recent statistics. 2, line 60: please state the three main subtypes. 2, line 63: “that include computed tomography (CT)…” 2, line 84: The authors state that nanoparticles have been investigated as imaging agents for GBM diagnosis. Have they been evaluated in clinical trials, or only in animal models? Please state explicitly. 2, line 92: “show advantage over…” 4, lines 151-153: The authors state that 5-ALA-based fluorescence-guided resection results in increased patient survival. Is this overall survival or progression-free survival? Please clarify. 4, line 164: “poor signal-to-noise ratio…” 5, line 207: The authors state that: “advances in material engineering and the development of nanotechnology revolutionized the way we diagnose and treat glioblastoma.” This is a very strong statement. Presently, the standard treatment for GBM is surgical resection followed by radiation and chemotherapy. GBM is diagnosed via standard imaging modalities (MRI, CT, PET, SPECT). To date nanotechnology has not had significant impact on the management of GBM patients. Therefore, I recommend re-writing the sentence to: “In general, advances in material engineering and the development of nanotechnology has the potential to revolutionize….” 5, line 216: The meaning of “20 rounds of exponential growth” is not clear. Do the authors mean 20 cell divisions? 5, line 229: “Their function contributes to sophisticated communication…” 7, line 286: “obtained CSF shortly after…” 7, line 316: The meaning of “or pulled randomly” is not clear. Please re-write for clarity. 9, line 375: “detectable alteration, most commonly…” 9, line 422: suggest eliminating “and others.” 9, line 424: “rather than the properties of the…” 10, line 443: “uptake of the drugs.” 10, line 452: “to cover the whole spectrum, but some interesting approaches are…” 11, line 491: “of the animals was shown…” 12, line 555: “three different polymeric nanoparticle carriers…” 13, line 583: “of monoclonal antibody…” 14, line 674: “Also, M1 prevalence in tumors…” 15, line 693: “which consists of a signaling peptide,…” 16, line 719: “Ultimately, 228 days after the disease recurred.” 16, line 722: “Application of CAR-T therapy…” 16, line 731: “Most attention has been focused on…” 16, line 739: “CT enabled tracking…” 17, line 751: “Subsequently, CAR-T…”

p. 19, line 851: “is biocompatible and is specifically…”

Reviewer 2 Report

This manuscript is a review covering important topics in nanomedicine and immunotherapy for glioblastoma. In general, the article is well-written, insightful and fairly comprehensive. I only have one suggestion for the authors’ consideration to further improve the quality of the manuscript. The authors should make a few tables to summarize most relevant studies and their major findings in each type of therapies, so the readers can find what they are interested more easily. Other than that, a well-done work.
